# Replacing Alfalfa with Paper Mulberry in Total Mixed Ration Silages: Effects on Ensiling Characteristics, Protein Degradation, and In Vitro Digestibility

**DOI:** 10.3390/ani11051273

**Published:** 2021-04-28

**Authors:** Rongrong Li, Mingli Zheng, Di Jiang, Pengjiao Tian, Menghu Zheng, Chuncheng Xu

**Affiliations:** 1College of Engineering, China Agricultural University, Beijing 100083, China; rongzuerr@163.com (R.L.); 13120018505@163.com (P.T.); zmheat@sina.com (M.Z.); 2Beijing Research and Development Center for Grass and Environment, Beijing Academy of Agriculture and Forestry Sciences, Beijing 100097, China; zhengml@cau.edu.cn; 3College of Food Science and Engineering, Henan University of Technology, Zhengzhou 450000, China; lychee_jiang@163.com

**Keywords:** paper mulberry, alfalfa, protein quality, in vitro digestibility, total mixed ration silage

## Abstract

**Simple Summary:**

The usage of alfalfa (*Medicago sativa* L.) as a dietary protein source for ruminants in China is limited by forage quality and planting scale. Paper mulberry (*Broussonetia papyrifera* L., RY) has emerged as a new and representative high-protein woody forage resource for ruminants. However, information is less available regarding how substituting RY for alfalfa affect the fermentation and protein quality in total mixed ration (TMR) silages. This study evaluated ensiling characteristics, protein quality, and in vitro digestibility in TMR silages by mixing RY with alfalfa at different rations. The TMR were made with alfalfa and RY mixtures (36.0%), maize meal (35.0%), oat grass (10.0%), soybean meal (7.5%), brewers’ grain (5.0%), wheat bran (5.0%), premix (1.0%), and salt (0.5%) on a dry matter basis, respectively. The alfalfa and RY mixtures were made in the following ratios of dry matter: 36:0 (RY0), 27:9 (RY9), 18:18 (RY18), 9:27 (RY27), and 0:36 (RY36). The results showed that RY substitution had no adverse effect on fermentation quality and nutritional composition, but inhibited true protein degradation, while decreasing in vitro dry matter and crude protein digestibility. Therefore, RY and alfalfa mixtures at a ratio of 18:18 is suitable for silage-based TMR.

**Abstract:**

To develop an alternative high-protein forage resource to alleviate ruminant feed shortages, we investigated the effects of replacing alfalfa (*Medicago sativa* L.) with different ratios of paper mulberry (*Broussonetia papyrifera* L., RY) on fermentation quality, protein degradation, and in vitro digestibility of total mixed ration (TMR) silage. The TMR were made with alfalfa and RY mixtures (36.0%), maize meal (35.0%), oat grass (10.0%), soybean meal (7.5%), brewers’ grain (5.0%), wheat bran (5.0%), premix (1.0%), and salt (0.5%) on a dry matter basis, respectively. The alfalfa and RY mixtures were made in the following ratios of dry matter: 36:0 (RY0), 27:9 (RY9), 18:18 (RY18), 9:27 (RY27), and 0:36 (RY36). After ensiling for 7, 14, 28, and 56 days, fermentation quality, protein degradation, and microbial counts were examined, and chemical composition and in vitro digestibility were analyzed after 56 days of ensiling. All TMR silages, irrespective of the substitution level of RY, were well preserved with low pH and ammonia nitrogen content, high lactic acid content, and undetectable butyric acid. After ensiling, the condensed tannin content for RY18 silages was higher than the control, but non-protein nitrogen, peptide nitrogen, and free amino acid nitrogen contents was lower, while the fraction B1 (buffer-soluble protein) was not different among all the silages. Dry matter and crude protein digestibility for RY27 and RY36 silages was lower than the control, but there was no difference between control and RY18 silages. This study suggested that ensiling RY with alfalfa inhibited true protein degradation, but decreased in vitro dry matter and crude protein digestibility of TMR silages, and that 18:18 is the optimal ratio.

## 1. Introduction

Alfalfa (*Medicago sativa* L.) is commonly used as an excellent high-protein forage for dairy cows, but the availability in China is limited by forage quality and planting scale. Since the quantity of alfalfa can no longer meet the requirements of dairy farms, the contradiction between demand and supply has increased sharply [1]. Finite access to high-quality forage decreases milk and meat production for livestock industry, undermines health and welfare of animals, and contributes to loss of economic benefits for farmers [2]. Therefore, exploring locally produced, high yielding, alternative forage resource to ensure satisfactory animal production becomes an urgent problem to be solved.

Paper mulberry (*Broussonetia papyrifera* L., RY), which is a deciduous tree of Moraceae family, has been widely cultivated as an emerging fodder tree in recent years, owing to its strong adaptability, fast growth, and high biomass yield. As one of China’s top ten targeted poverty alleviation projects, over 300,000 hectares of RY are cultivated in China [3]. It is rich in crude protein (CP) and contains highly digestible fiber and minerals [2]. Multiple biologically active ingredients are also associated with its antioxidant and anti-inflammatory functions in ruminants [3,4]. These merits make it attractive to exhibit great potential and applicability as a nutrient-enriched forage resource for local animal production.

Dietary inclusion of RY provides an alternative approach to alleviate feed shortages for livestock production [5]. However, the growing season of RY is majorly concentrated during the summer rainy season in China, which impairs its sustainable application in dairy rations for all years. Over-production would be discarded in the field, causing squandering of resources [6]. Recently, the conservation of RY as silage has proven to be effective for overcoming the gap between annual animal production and seasonal imbalance [2]. Nevertheless, similar to other high-protein forages (e.g., legume forages), the fermentation quality of ensiled RY is usually unsatisfactory, as indicated by extensive proteolysis, high butyric acid content, and unfavorable odor [6,7]. During the protein degradation process, non-protein nitrogen (NPN), including mainly peptides, free amino acids (FAA), and ammonia nitrogen (NH_3_-N), is largely produced and accompanied by the change in true protein fraction and digestibility of the remaining protein [8]. Extensive proteolysis leads to the low nitrogen utilization by ruminants, thus, resulting in large economic losses to farmers and adverse environmental issues.

Total mixed ration (TMR) silage has been proposed as an ideal method that does not only help develop new feed resources and balance the moisture content of moist crops, but also improves palatability by altering the odors and flavor of feed resources and preserves nutritional value [9]. Therefore, the utilization of TMR silage may be a preferred method for efficient utilization of RY. The purpose of this study was to evaluate the effects of substituting different levels of RY for alfalfa on the fermentation quality, chemical composition, protein degradation, and in vitro digestibility of TMR silages.

## 2. Materials and Methods

### 2.1. Plant Material and Silage Preparation

Hybrid RY (Huagou 101) and alfalfa (WL656HQ) were cultivated with no herbicide or fertilizer application at the Zhuozhou Experimental Station (115°51′46″ N, 39°28′30″ E) of China Agricultural University. Three fields were obtained randomly from various forage fields, with three replicates of fresh RY and alfalfa. The whole foliage of RY was harvested, with a sickle at 40 cm stubble left at an approximate height of 1.2–1.5 m. Fourth-cut alfalfa was mowed with 10 cm of stubble left at the squaring stage of maturity. Three replicates of forages were chopped separately with a hand hay cutter into 2–3 cm pieces and mixed thoroughly by hand before adding other ingredients.

The TMR were made with alfalfa and RY mixtures (36.0%), maize meal (35.0%), oat grass (10.0%), soybean meal (7.5%), brewers’ grain (5.0%), wheat bran (5.0%), premix (1.0%), and salt (0.5%) on a dry matter basis, respectively. The alfalfa and RY mixtures were made in the following ratios of dry matter: 36:0 (RY0), 27:9 (RY9), 18:18 (RY18), 9:27 (RY27), and 0:36 (RY36). The experimental diets were formulated according to a Chinese Feeding Standard for Dairy Cattle [10]. After mixing, approximately 400 g of the material was packed into plastic vacuum packing bag silos and sealed with a vacuum packing machine (BH950, Matsushita, Osaka, Japan), and stored at an ambient temperature (15–22 °C). Three silos from each treatment were opened after 7, 14, 28, and 56 days of ensiling, respectively.

### 2.2. Fermentation Quality, Chemical and Microbial Analysis

When each silo was opened, silage samples were mixed thoroughly and divided into three portions for fermentation characteristics, chemical composition, and microbial analysis. One of them was dried by an air oven at 65 °C for 48 h to determine dry matter (DM) content. Total nitrogen (TN) was measured using the Kjeldahl nitrogen analyzer (KDY-9830) and CP content was calculated by Kjeldahl nitrogen multiplied by 6.25. Residual moisture was corrected by drying at 105 °C for an additional 2 h, and chemical composition and in vitro digestibility were determined relative to DM content. The contents of ether extract and water soluble carbohydrates (WSC) were measured as described by AOAC [11] and Owens et al. [12], respectively. Neutral detergent fiber (NDF), acid detergent fiber (ADF), and acid detergent lignin (ADL) were analyzed according to the procedures of Van Soest et al. [13]. Buffering capacity was measured using the method of Zhang et al. [7]. In terms of tannin profile, total phenols and simple phenols were determined using the Folin-Ciocalteu colorimetry, according to the method of Makkar [14]. Hydrolysable tannins content was calculated by subtracting simple phenols from total phenols and condensed tannins contents were determined according to Coblentz and Grabber [15].

For microbial counts analysis, wet silage (20 g) was diluted with 180 mL of sterilized distilled water and then serial dilutions (10^−1^–10^−7^) were prepared using sterilized water. Lactic acid bacteria counts were measured using a plate of de Man Rogosa and Sharpe (MRS) agar (Difco Laboratories, Detroit, MI, USA) incubated at 37 °C for 48 h under anaerobic conditions. Aerobic bacteria and yeast, and mould were enumerated on nutrient agar (Nissui, Tokyo, Japan), Potato Dextrose Agar (Nissui, Tokyo, Japan), and Rose Bengal agar (Nissui, Tokyo, Japan) after incubation for 48 h at 30 °C under aerobic conditions.

To determine the pH and organic acid contents, the remaining wet silage (10 g) was diluted with 90 mL of sterilized distilled water and then filtered with four layers of medical gauze and a qualitative filter paper. The pH was tested with an electrode pH meter (S20K, Mettler Toledo, Greifensee, Switzerland). The contents of organic acids, including lactic, acetic, propionic, and butyric acids, were analyzed by HPLC (LC-10A, Shimadzu, Tokyo, Japan). The parameters were as follows: column, Shodex RS PAK KC-811S-DVB; mobile phase, 3 mmol/L HClO_4_; flow rate, 1.0 mL/min; and oven temperature, 50 °C. The V-Score was calculated to evaluate the fermentation quality of TMR silages [16].

### 2.3. Protein Degradation Indicator Analysis

Triplicate silos for each treatment were opened after 7, 14, 28, and 56 days of ensiling. Protein fractions were calculated according to Cornell Net Carbohydrate and Protein System (CNCPS) [17]. In this system, the fraction A is defined as NPN, which is calculated by subtracting the value of true protein nitrogen precipitated with trichloroacetic acid from total nitrogen. The fraction B is degradable true protein, which is further subdivided into B1 (rapidly), B2 (intermediately), and B3 (slowly) degradable fractions. Fraction C is considered to be indigestible and is insoluble in acid detergent solution. The true protein nitrogen, buffer-insoluble protein nitrogen, neutral detergent-insoluble nitrogen, and acid detergent-insoluble nitrogen of samples were analyzed as described by Licitra et al. [18].

For nitrogen distribution analysis, contents of NH_3_-N and free amino acid nitrogen (FAA-N) were determined according to Broderick and Kang [19]. Peptide nitrogen (peptide-N) content was analyzed by the increase in FAA-N in the trichloroacetic acid after digesting with 6 mol/L HCl for 21 h at 105 °C under an N_2_ atomosphere [20].

### 2.4. In Vitro Digestibility Analysis

The in vitro digestibility trail was performed according to the procedure described by Tilley [21] with some modifications. Fresh ruminal liquid was collected from three rumen-fistulated Angus bullocks (600 ± 1.6 kg of live weight) 2 h before the morning feeding. The Angus bullocks were fed with a diet containing 8% soybean residue, 20% wheat straw, 22% corn silage, and 50% concentrate on a DM basis with free access to water. The rumen fluid was quickly filtered, mixed with a buffer solution, and kept at 39 °C in a water bath under continuous flushing with CO_2._ The artificial buffer solution was prepared as described by Menke et al. [22]. Briefly, oven-dried samples were placed in filter bags that were washed with acetone and dried to a constant weight. Each bag was heat-sealed and put into a separate preheated serum bottle with rumen fluid-buffer mixture under CO_2_ at 39 °C. Each sample was set at three parallels and at another six blanks, and the entire experiment was repeated in three runs. After a period of 48 h incubation under CO_2_ at 39 °C, the rumen fluid-buffer mixture of each serum bottle was discarded immediately, and freshly-made pepsin solution was added to each serum bottle with a washed filter bag. After another 48 h of the incubation, all the filter bags were washed with distilled water, and then dried to a constant weight at 65 °C. Digestibility of DM (DMD), CP (CPD), and NDF (NDFD) were determined based on the differences in their respective weights before and after incubation. All procedures in this experiment were approved by the Institutional Animal Care and Use Committee of China Agricultural University (DK 996 No. 20144-2), Beijing, China.

### 2.5. Statistical Analysis

The experiment was a completely randomized design. Prior to statistical analysis, all data are subjected to a homogeneity test of variance. The normality was tested with the Shapiro–Wilk test. All data can be analyzed by ANOVA.Microbial data were log_10-_transfomed and presented on a fresh matter basis prior to statistical analysis. The effects of treatment, storage days, and their interaction on fermentation quality, nitrogen constribution, protein fraction, and microbial counts of samples were analyzed using a two-way ANOVA by using GLM procedure of SPSS (version 21.0, IBM Corp, Armonk, NY, USA). The statistics model used was [23]:Y_ij_ = μ + I_i_ + T_j_ + (I × T)_ij_ + e_ij_(1)
where Y_ij_ was the response variable, μ was the overall mean, I_i_ was the treatment effect, T_j_ was the effect of the storage period, (I × T)_ij_ was the effect of interaction between the treatment and the storage period, and e_ij_ was the residual error. The statistical difference between means was analyzed by using Tukey’s multiple comparison and considered significant at *p* < 0.05.

Chemical and microbial compositions data of pre-ensiled materials, chemical composition, and in vitro digestibility data on day 56 were subjected to one-way ANOVA. The statistics model used was:Y _i_ = μ + I_i_ + e_i_(2)
where Y _i_ was the response variable, μ was the overall mean, I_i_ was the treatment effect, and e_i_ was the residual error. Orthogonal polynomial contrasts were used to determine the linear and quadratic effects of the treatments. *p* < 0.05 was considered statistically significant.

## 3. Results

### 3.1. Characteristics of Pre-Ensiled Materials

Paper mulberry and alfalfa had low DM contents, namely 216 and 242 g/kg, respectively (Table 1). Paper mulberry contained higher CP and WSC contents (*p* < 0.05) and lower NDF and ADF contents than alfalfa (*p* < 0.05). Condensed tannin and hydrolysable tannin contents were high in RY, but were undetectable in alfalfa. Lactic acid bacteria, yeast, and aerobic bacteria counts were higher in RY than in alfalfa (*p* < 0.05). 

Chemical and microbial compositions were affected by the treatments (*p* < 0.05), except for the contents of ADL, WSC, and ether extract and the numbers of yeast and mould (Table 2). The CP content was between 152 and 165 g/kg DM among the five treatments, and RY36 had higher CP content than RY0 and RY9 (*p* < 0.05). The NDF and ADF contents linearly decreased (*p* < 0.05) and DM quadratically decreased (*p* < 0.05), whereas buffer capacity linearly increased (*p* < 0.05) and condensed tannin and hydrolysable tannin quadratically increased (*p* < 0.05) with an increasing proportion of RY in pre-ensiled TMR. Lactic acid bacteria count linearly increased (*p* < 0.05) with an increasing proportion of RY in pre-ensiled TMR, with an average value of 7.04 to 7.88 log_10_ cfu/g FM across the five treatments.

### 3.2. Dynamic Changes in Fermentation Quality of TMR Silages

Treatment, storage days, and their interactions significantly affected silage pH and organic acid contents (*p* < 0.05) (Table 3). Silage pH increased continuously with an increasing proportion of RY in the TMR silages, and RY36 had higher pH than RY0 during ensiling (*p* < 0.05). As the ensiling period increased, the pH in all TMR silages gradually decreased, with the pH ranging from 4.31 to 4.40 at 56 days. As the proportion of RY increased, lactic acid and acetic acid contents continuously increased in all TMR silages, and RY36 had higher lactic acid and acetic acid contents than RY0 and RY9 during ensiling (*p* < 0.05). Butyric acid was undetectable in all TMR silages.

### 3.3. Dynamic Changes in Microbial Counts of TMR Silages

Treatment and storage days significantly affected lactic acid bacteria counts (*p* < 0.05), but did not affect aerobic bacteria counts (*p* > 0.05) (Table 4). Lactic acid bacteria count in RY36 were higher than those in RY0 during the first 14 days of ensiling (*p* < 0.05). Yeast and mould counts were under detectable level (< 2.40 log_10_ cfu/g FM).

### 3.4. Chemical Composition and V-Score of 56-Day TMR Silages

Increasing the proportion of RY in TMR silages linearly decreased DM content, but linearly increased CP and hydrolysable tannin content (*p* < 0.05), which did not affect ADF and ADL contents (Table 5) (*p* > 0.05). The buffer capacity of all TMR silages ranged from 41.3 to 58.8 g LA/kg DM, and the lowest value was observed in RY0. Condensed tannin content increased significantly with an increasing proportion of RY in TMR silages, with average values ranging from 0.52 to 7.32 g/kg DM across the five treatments. The five TMR silages showed high V-Score ranging from 90.31 to 95.43 on day 56.

### 3.5. Dynamic Changes in Nitrogen Distribution of TMR Silages

Treatment and storage days significantly affected NPN, peptide-N, FAA-N, and NH_3_-N contents of TMR silages (*p* < 0.05) (Table 6). A sharp increase in the NPN content in all TMR silages was observed during the first 7 days of ensiling (*p* < 0.05). Afterward, it continued increasing, but a slow pace, until 14 days, in all silages (*p* < 0.05), and then remained almost constant in all silages except for RY0 (*p* > 0.05). After 56 days of ensiling, RY18, RY27, and RY36 had lower NPN contents than RY0 (*p* < 0.05). A substantial increase in FAA-N content in RY0, RY9, and RY36 silages after ensiling were observed (*p* < 0.05) and RY36 silages had lower FAA-N content than RY0 and RY9 silages (*p* < 0.05). The NH_3_-N content in all TMR silages continued to increase after ensiling, and RY36 silages had higher NH_3_-N content than RY0 throughout the entire ensiling process (*p* < 0.05).

### 3.6. Dynamic Changes in Protein Fraction of TMR Silages

Treatments and storage days significantly influenced the proportion of A, B1, B2, and true protein fractions of TMR silages (*p* < 0.05) (Table 7). The proportion of true protein continuously decreased throughout the entire ensiling process and the fastest decline rate was recorded during the first 7 days. After 56 days of ensiling, RY27 and RY36 had higher true protein content than RY0 (*p* < 0.05). Fraction B1 in RY36 was higher than that in RY0 before ensiling (*p* < 0.05). However, no difference was observed between them after ensiling. Increasing the proportion of RY linearly increased the fraction B2 proportion and RY36 had a higher fraction B2 than RY0 and RY9 on day 56 (*p* < 0.05). Treatment had no effect on fraction B3 and fraction C (*p* > 0.05), whereas there was a decreasing trend in fraction C with increasing levels of RY in TMR silages (0.05 < *p* < 0.1). Fraction C continued at a relatively stable level after an increase on day 56, with a range of 47.5 to 52.3 g/kg TN (*p* < 0.05).

### 3.7. In Vitro Digestibility of 56-Day TMR Silages

The DM and CP digestibility linearly decreased (*p* < 0.05) with increasing the proportion of RY in TMR silages. The lower DM and CP digestibility were observed in RY27 and RY36 than RY0 (*p* < 0.05) (Table 8).

## 4. Discussion

The chemical composition of RY and alfalfa used in this study was similar to the values reported for RY and alfalfa harvested at the juvenile and squaring stages, respectively, in Northern China [7,24]. The CP and WSC contents were much higher in RY of raw material than those of alfalfa, indicating that the RY may be used as a superior high-protein forage resource to supplement ruminant feed. Sole RY is difficult to ensile successfully owing to its high buffer capacity and CP content as well as low DM contents. Thus, there is a need to formulate RY with other ingredients as a TMR silage. The purpose of this study was to determine the effects of replacing alfalfa with different levels of RY on fermentation quality, chemical composition, protein degradation, and in vitro digestibility in TMR silages.

After ensiling for 56 days, all TMR silages irrespective of the substitution of RY were well fermented with dominant lactic acid content, low pH, NH_3_-N content, and undetectable butyric acid content. This result is consistent with that in our previous studies [25,26], which suggested that high-moisture crops can be well-preserved by formulating TMR silage. This may be related to the sufficient WSC and appropriate moisture contents in all samples to meet the conditions for obtaining satisfactory silage quality. The addition of RY increased the lactic acid bacteria load of pre-ensiled TMR, leading to a higher lactic acid content of silages in RY36 relative to RY0 during ensiling. However, consistent with the elevated lactic acid levels, the pH value increased in response to RY addition. This result was mainly related to their high buffering capacity, which impeded the decline in pH during the ensiling process. In that case, substrate consumption in RY36 could not be effectively suppressed by microbial activity. Therefore, in contrast to other treatments, RY36 accumulated more NH_3_-N and reserved less WSC. Tian et al. [27] reported that high pH and NH_3_-N content in TMR silages resulted from high buffering capacity, even with a high lactic acid content.

Treatment differences in the chemical composition of the TMR silages generally reflected compositional variations in the forage materials and their mixing proportions before ensiling. In this study, the higher CP and lower NDF in RY27 and RY36 silages relative to RY0 after ensiling was mainly attributed to the high CP and low NDF in RY materials. Previous research found that forage chemical composition can affect animal feeding behavior, DM intake, metabolism, and performance [28,29]. Feed fiber is conducive to rumination and rumen pH. Consequently, it is negatively correlated with DM intake and digestibility [30]. With the higher CP and lower NDF, an addition level of RY in TMR silages possessed a higher nutritive value theoretically, which could be preferentially recommended to feed the high-lactation dairy cows. However, the higher condensed tannin and hydrolysis tannin in TMR silages with an increasing level of RY were also detected in this study. Tannin in diets generally has harmful effects on nutrient availability in ruminants [31]. Thus, a digestion trial is needed to further evaluate the feed value of these TMR silages. The ensiling process apparently changed the chemical composition of all the TMR silages. In comparison to the pre-ensiled TMR, the DM content in all TMR silages constantly decreased with ensiling. The readily degradable components (e.g., WSC) of silages were transferred into organic acids, ethanol, and carbon dioxide by microorganisms when ensiling fermentation. The CP content increased throughout the experiment after ensiling, indicating efficient fermentation and nutritional preservation of all silages. Similarly, Chen et al. [32] observed an increase in CP content in TMR after ensiling, which was attributed to a concentration effect and microbial crude protein. Compared with pre-ensiled TMR, the ether extract, ADF, and ADL contents increased after ensiling likely as a result of being represented on the basis of DM. In contrast, NDF content decreased slightly in all silages after fermentation in this study. Yin et al. [33] ascribed this increase to acid hydrolysis of cell wall fractions during silage fermentation. Overall, substituting RY with alfalfa in TMR silages did not have adverse effects on ether extract content, but effectively increased CP content and decreased NDF content over the control silages, suggesting an improved nutritive composition of TMR silages by introducing RY.

In terms of forage crops, especially for high-protein herbages, it is inevitable that large quantities of true protein are converted into NPN during ensiling, resulting in poor nitrogen utilization for ruminants. In the present study, NPN content in all silages increased rapidly during the first 7 days of ensiling, as supported by Hao et al. [34] in TMR silages. However, higher NPN content with lower silage pH in RY36 compared with RY0 is in agreement with Sousa et al. [35], who reported that protein breakdown during ensiling can be reduced by a rapid decrease in silage pH. Earlier studies have shown that the protein degradation process during ensiling of tannin-containing species, such as purple prairie clover (*Dalea purpurea*) and sainfoin (*Onobrychis viciifolia*), is suppressed in comparison to that of non-tannin containing forages, such as alfalfa [36]. Thus, the inhibition of protein degradation in RY36 during silage fermentation in this study is likely attributed to its highly condensed tannin content. Protein degradation in ensiled forages proceeds through two major pathways. True protein is hydrolyzed to peptides and FAA is hydrolyzed mainly by plant proteases under aerobic conditions during the initial ensiling process, and mainly by microbial enzymes when the FAA is deaminated to produce ammonia during anaerobic fermentation [35]. In this study, the initial increases in peptide-N content in all TMR silages during the first 14 days indicated that protein hydrolysis initially exceeded deamination by microbes while the contribution of microbes became greater than plant protease during the later stage of ensiling. This explains the gradually decreased peptide-N content after 14 days. Free amino acid nitrogen content is closely related to the extent of proteolysis [37]. The lower FAA-N content in RY27 and RY36 silages showed that the hydrolysis of true protein and peptides was weaker in RY27 and RY36 silages than control silages. Li et al. [8] reported that peptides can be protected from degradation by bonding to tannin, thus, inhibiting the production of FAA. Notably, increasing the level of RY did not reduce deamination of amino acids in the present study. Instead, the NH_3_-N content in RY36 was even higher than that in RY0 at every phase of ensiling. Guo et al. [37] found that tannin additives can lower the formation of NH_3_-N only when its application rate exceeds 35 g/kg DM. This means that only high levels of tannin may have caused the inhibition of deamination. Balanced amino acid composition of the diet is essential for maximizing nutrient utilization and animal productivity. Lysine and methionine play an important role in the metabolism in organisms and are the first two limiting amino acids necessary for dairy cows [38]. Further quantitative evaluation of amino acid degradation in TMR silages is essential to add more information on accurate animal feeding and production when RY is introduced. Overall, increasing a substituted level of RY in TMR silages decreased NPN, peptide-N, and FAA-N content, thereby effectively protecting true protein from hydrolysis. Yet, it did not prevent the deamination of FAA because of the increasing NH_3_-N content.

The ensiling process did not affect the proportion of the B3 fraction, but increased the proportions of A and C fractions, while decreasing the proportions of true protein as well as B1 and B2 fractions, suggesting that the ensiling process not only negatively facilitated proteolysis, but also decreased the protein quality of TMR silages by converting true protein into soluble NPN. The higher fraction C in all TMR silages than fresh materials might mainly result from the heat accumulation owing to the Maillard reaction, as previously reported by Guo et al. [38]. It should be noted that the shifts within the fraction B1 from 0 to 56 days showed differentiation among different treatments. There was a declining variation in fraction B1 before and after ensiling with an increasing proportion of RY in TMR silages. This indicated that the addition of RY to TMR silages minimized the degree of proteolysis and prevented the degradation of soluble true protein into NPN. Increasing the proportion of RY increased the true protein content in TMR silages, which was mainly associated with the increase in fraction B2 content. According to the CNCPS, fraction B2 is fermented only at a degradation rate of 50–150 g/kg h, and a large amount escaped to the lower gut [17]. It can be predicted that the amount of true protein flowing to the lower gut increases with an increased proportion of RY based on fraction B2 data. Fraction C contains lignin, tannin-protein complexes, and Maillard reaction products that are indigestible in the rumen and intestine [17]. The addition of RY did not affect the proportions of C fractions, but increased the B2 fraction and true protein content, while decreasing the non-protein nitrogen content of TMR silages, implying a potential beneficial effect from RY and leading to higher protein quality than that of the control silages.

A nutritional value of feeds is primarily determined by digestibility for the ruminant and an in vitro culture has been developed as a common technique owing to its high correlation with in vivo digestibility and convenient operation [21]. Digestibility mainly depends on the chemical composition of forages, and high CP and low plant cell wall contents are beneficial for improving DM digestibility. However, increasing the proportion of RY in TMR silages resulted in an increase in CP content and a decrease in NDF content and DM digestibility in the TMR silages in the present study, which might be partly attributed to the high tannin content of RY. Cummins [39] observed negative effects of tannin content on in vitro DM digestibility of a high-tannin hybrid of sorghum compared to a low-tannin hybrid. The decreased CP digestibility in TMR silages with an increased level of RY was a result of the composition of CP of RY being less digestible than that of alfalfa, which was reflected in the lower soluble CP and greater tannin contents in TMR silages containing higher level of PM, resulting in a reduction of total degradable CP fractions. Xu et al. [9] used Suffolk Wethers to compare the digestibility of TMR silages containing different levels of green tea grounds rich in tannin and found that the DM and CP digestibility of silages with 150 g/kg concentrations of green tea grounds were lower than those with 0 and 50 g/kg green tea grounds.

The study of Si et al. [3] indicated that substituting 10% and 15% of RY silage for a portion of whole corn silage, alfalfa hay, and oat hay improved milk quality, and immunity and antioxidant activity of dairy cows. Tao et al. [4] reported that diets supplemented with 15% RY silage improved the meat quality and growth performance of beef cattle. This study indicates that the addition of 18% RY was the most optimal for formulating TMR silage because of the balance of fermentation quality, feed-nutritional value, protein degradation, and in vitro digestibility. Future research is needed to investigate intake and production, particularly nitrogen efficiency and urinary nitrogen excretion, when mixed TMR silage is fed to dairy cows.

## 5. Conclusions

The results of the present study revealed that RY could serve as a potential protein feed source of TMR silage to replace a portion of alfalfa and five mixtures of TMR silages in which RY was substituted for alfalfa were well preserved with high fermentation quality and nutrient composition. After ensiling, the NPN and peptide-N contents for RY18 silages were lower than that of the control, but the B1, B3, and C fractions were no different among all the silages. Digestibility of DM and CP for RY27 and RY36 silages was lower than the control, but there was no difference between the control and RY18 silages. Therefore, it appears that TMR silages produced an RY to alfalfa ratio of 18:18, which was suitable due to balancing the silage quality, protein degradation, and in vitro digestibility.

## Figures and Tables

**Table 1 animals-11-01273-t001:** Chemical composition (g/kg DM) and microbial counts (log_10_ cfu/g FM) of paper mulberry and alfalfa materials.

Items	Paper Mulberry	Alfalfa	SEM	*p*-Value
Chemical composition				
DM (g/kg FM)	216 ^b^	242 ^a^	0.341	0.003
CP	235 ^a^	204 ^b^	0.300	<0.001
NDF	331 ^b^	402 ^a^	0.384	<0.001
ADF	196 ^b^	242 ^a^	0.166	0.016
ADL	40.9 ^b^	51.4 ^a^	0.089	<0.001
Buffer capacity (g LA/kg DM)	97.6 ^a^	62.1 ^b^	0.051	<0.001
WSC	74.9 ^a^	41.5 ^b^	0.074	<0.001
Condensed tannin	18.5	ND	-	-
Hydrolysable tannin	7.10	ND	-	-
Microbial composition				
Lactic acid bacteria	5.38 ^a^	4.16 ^b^	0.048	<0.001
Aerobic bacteria	6.96 ^a^	5.77 ^b^	0.069	<0.001
Yeast	4.11 ^a^	3.49 ^b^	0.114	0.030
Mould	4.82	4.77	0.092	0.171

Values in the same row (a,b) with different superscripts are significantly different (*p* < 0.05). DM, dry matter. FM, fresh matter. LA, lactic acid. ND, not detected. CP, crude protein. NDF, neutral detergent fiber. ADF, acid detergent fiber. ADL, acid detergent lignin. WSC, water soluble carbohydrate. SEM, standard error of means. N = 3.

**Table 2 animals-11-01273-t002:** Effect of paper mulberry replacement levels on chemical composition (g/kg DM) and microbial counts (log_10_ cfu/g FM) of pre-ensiled TMR.

Items	Treatments					SEM	*p*-Value		
	RY0	RY9	RY18	RY27	RY36		T	L	Q
Chemical composition									
DM (g/kg FM)	439 ^a^	437 ^ab^	434 ^abc^	431 ^bc^	428 ^c^	0.117	0.039	0.007	0.015
CP	152 ^c^	155 ^bc^	158 ^abc^	161 ^ab^	165 ^a^	0.121	0.009	0.004	0.033
WSC	122	124	127	131	134	0.269	0.064	0.255	0.865
NDF	324 ^a^	316 ^ab^	307 ^abc^	299 ^bc^	292 ^c^	0.464	0.022	0.005	0.645
ADF	164 ^a^	159 ^ab^	155 ^ab^	151 ^ab^	147 ^b^	0.367	0.037	0.010	0.541
ADL	49.1	47.0	46.5	46.3	45.1	0.064	0.445	0.094	0.656
Ether extract	35.3	35.9	36.7	37.5	38.2	0.380	0.131	0.307	0.469
Buffer capacity (g LA/kg DM)	66.1 ^c^	70.3 ^bc^	75.0 ^abc^	80.2 ^ab^	84.8 ^a^	1.864	<0.001	<0.001	0.173
Condensed tannin	0.33 ^e^	1.99 ^d^	3.76 ^c^	5.45 ^b^	7.07 ^a^	0.076	<0.001	<0.001	0.016
Hydrolysable tannin	0.19 ^d^	0.81 ^cd^	1.59 ^bc^	2.26 ^ab^	2.90 ^a^	0.041	<0.001	0.011	0.030
Microbial counts									
Lactic acid bacteria	7.04 ^c^	7.23 ^bc^	7.50 ^ab^	7.74 ^a^	7.88 ^a^	0.056	<0.001	0.022	0.176
Aerobic bacteria	5.99 ^c^	6.19 ^bc^	6.33 ^abc^	6.56 ^ab^	6.62 ^a^	0.025	<0.001	<0.001	0.166
Yeast	6.01	5.85	5.89	5.90	5.86	0.025	0.225	0.139	0.278
Mould	5.88	5.86	5.94	5.67	5.68	0.041	0.355	0.090	0.458

Values in the same row (a–e) with different superscripts are significantly different (*p* < 0.05). T, treatment effects. L, linear effects. Q, quadratic effects. FM, fresh matter. DM, dry matter. CP, crude protein. WSC, water soluble carbohydrates. NDF, neutral detergent fiber. ADF, acid detergent fiber. ADL, acid detergent lignin. LA, lactic acid. SEM, standard error of means. N = 3. RY0, 0% RY + 36% alfalfa. RY9, 9% RY + 27% alfalfa. RY18, 18% RY + 18% alfalfa. RY27, 27% RY + 9% alfalfa. RY36, 36% RY + 0% alfalfa.

**Table 3 animals-11-01273-t003:** Effect of paper mulberry replacement levels and storage days on fermentation quality of TMR silages (g/kg DM).

Items	Treatment	Storage Days				SEM	*p*-Value		
		7	14	28	56		T	S	T×S
pH	RY0	4.53 ^bA^	4.41 ^cB^	4.36 ^bBC^	4.31 ^bC^	0.073	0.004	<0.001	0.022
	RY9	4.55 ^abA^	4.43 ^cB^	4.39 ^abBC^	4.34 ^abC^				
	RY18	4.59 ^abA^	4.48 ^bcB^	4.40 ^abBC^	4.37 ^abC^				
	RY27	4.65 ^aA^	4.54 ^abB^	4.42 ^abC^	4.38 ^abC^				
	RY36	4.66 ^aA^	4.58 ^aB^	4.44 ^aC^	4.40 ^aC^				
Lactic acid	RY0	38.9 ^cC^	44.2 ^cB^	49.8 ^cA^	52.9 ^cA^	0.081	<0.001	<0.001	0.019
	RY9	41.2 ^bcB^	46.0 ^bcB^	52.9 ^bcA^	53.6 ^cA^				
	RY18	43.0 ^abcC^	48.3 ^abcBC^	53.6 ^abcAB^	55.9 ^bcA^				
	RY27	44.6 ^abC^	51.0 ^abB^	56.7 ^abA^	59.0 ^abA^				
	RY36	46.7 ^aC^	52.5 ^aB^	58.9 ^aA^	60.2 ^aA^				
Acetic acid	RY0	3.81 ^cC^	4.13 ^cC^	5.30 ^cB^	6.42 ^dA^	0.027	<0.001	<0.001	<0.001
	RY9	4.30 ^cB^	4.99 ^cB^	7.21 ^bA^	8.21 ^cA^				
	RY18	5.23 ^bC^	6.27 ^bB^	7.35 ^bB^	9.10 ^bcA^				
	RY27	7.62 ^aC^	7.77 ^aC^	9.00 ^aB^	10.5 ^abA^				
	RY36	7.73 ^aC^	8.27 ^aC^	9.40 ^aB^	11.0 ^aA^				
Propionic acid	RY0	0.43 ^cB^	0.73 ^cB^	1.08 ^dAB^	1.67 ^bA^	0.012	<0.001	<0.001	0.003
	RY9	0.50 ^cB^	0.85 ^cB^	1.56 ^cdA^	2.11 ^bA^				
	RY18	0.96 ^bcC^	1.34 ^bcBC^	1.80 ^bcB^	3.06 ^aA^				
	RY27	1.44 ^abC^	2.06 ^abBC^	2.35 ^abB^	3.30 ^aA^				
	RY36	1.87 ^aC^	2.65 ^aB^	2.81 ^aAB^	3.42 ^aA^				

Values in the same column (a–d) or in the same row (A–D) with different superscripts are significantly different (*p* < 0.05). DM, dry matter. SEM, standard error of means. N = 3. T, effect of treatment. S, effect of storage days. T × S, effect of treatment and storage days interactions. RY0, 0% RY + 36% alfalfa. RY9, 9% RY + 27% alfalfa. RY18, 18% RY + 18% alfalfa. RY27, 27% RY + 9% alfalfa. RY36, 36% RY + 0% alfalfa.

**Table 4 animals-11-01273-t004:** Effect of paper mulberry replacement levels and storage days on microbial counts of TMR silages (log_10_ cfu/g FM).

Items	Treatment	Storage Days				SEM	*p*-Value		
		7	14	28	56		T	S	T × S
Lactic acid bacteria	RY0	8.36 ^bA^	8.20 ^bA^	8.14 ^A^	7.49 ^B^	0.105	0.034	0.041	0.088
	RY9	8.48 ^bA^	8.30 ^abA^	7.97 ^AB^	7.49 ^B^				
	RY18	8.68 ^abA^	8.21 ^bAB^	8.03 ^BC^	7.58 ^C^				
	RY27	9.04 ^aA^	8.51 ^abAB^	8.35 ^BC^	7.69 ^C^				
	RY36	8.99 ^aA^	8.73 ^aAB^	8.39 ^B^	7.73 ^C^				
Aerobic bacteria	RY0	4.34	4.29	4.17	4.05	0.120	0.230	0.447	0.918
	RY9	4.36	4.25	4.20	4.10				
	RY18	4.38	4.31	4.19	4.17				
	RY27	4.40	4.38	4.24	4.17				
	RY36	4.57	4.43	4.31	4.29				

Values in the same column (a–c) or in the same row (A–C) with different superscripts are significantly different (*p* < 0.05). FM, fresh matter. SEM, standard error of means. N = 3. T, effect of treatment. S, effect of storage days. T × S, effect of treatment and storage days interactions. RY0, 0% RY + 36% alfalfa. RY9, 9% RY + 27% alfalfa. RY18, 18% RY + 18% alfalfa. RY27, 27% RY + 9% alfalfa. RY36, 36% RY + 0% alfalfa.

**Table 5 animals-11-01273-t005:** Effect of paper mulberry replacement levels on chemical composition and V-Score of 56-day TMR silages (g/kg DM).

Items	Treatments					SEM	*p*-Value		
	RY0	RY9	RY18	RY27	RY36		T	L	Q
DM (g/kg FM)	432 ^a^	429 ^ab^	426 ^abc^	422 ^bc^	418 ^c^	0.577	0.015	0.020	0.227
CP	159 ^c^	162 ^bc^	165 ^abc^	168 ^ab^	172 ^a^	0.126	0.036	0.014	0.259
Ether extract	36.1	36.8	37.6	38.5	39.4	0.461	0.255	0.491	0.605
WSC	47.6 ^a^	43.3 ^ab^	35.9 ^abc^	30.1 ^bc^	24.5 ^c^	0.037	0.019	0.022	0.025
NDF	305 ^a^	296 ^ab^	287 ^abc^	278 ^bc^	271 ^c^	1.274	0.021	0.044	0.292
ADF	170	169	166	164	160	0.448	0.119	0.051	0.156
ADL	49.4	47.8	47.0	47.2	46.3	0.079	0.414	0.194	0.240
Buffer capacity (g LA/kg DM)	41.3 ^b^	45.8 ^ab^	50.5 ^ab^	54.6 ^ab^	58.8 ^a^	1.142	0.031	0.022	0.191
Condensed tannin	0.52 ^e^	2.10 ^d^	3.99 ^c^	5.71 ^b^	7.32 ^a^	0.055	0.002	<0.001	0.026
Hydrolysable tannin	0.30 ^d^	0.92 ^cd^	1.64 ^bc^	2.32 ^ab^	3.01 ^a^	0.029	0.006	0.015	0.044
V-Score	95.43	93.65	92.19	90.90	90.31	0.511	0.054	0.131	0.166

Values in the same row (a–e) with different superscripts are significantly different (*p* < 0.05). DM, dry matter. FM, fresh matter. CP, crude protein. WSC, water soluble carbohydrate. NDF, neutral detergent fiber. ADF, acid detergent fiber. ADL, acid detergent lignin. LA, lactic acid. SEM, standard error of means. N = 3. T, treatment effects. L, linear effects. Q, quadratic effects. RY0, 0% RY + 36% alfalfa. RY9, 9% RY + 27% alfalfa. RY18, 18% RY + 18% alfalfa. RY27, 27% RY + 9% alfalfa. RY36, 36% RY + 0% alfalfa.

**Table 6 animals-11-01273-t006:** Effect of paper mulberry replacement levels and storage days on nitrogen distribution of TMR silages (g/kg TN).

Items	Treatment	Storage Days					SEM	*p*-Value		
		0	7	14	28	56		T	S	T × S
NPN	RY0	208 ^aD^	392 ^aC^	438 ^aB^	463 ^aAB^	487 ^aA^	1.147	<0.001	<0.001	0.694
	RY9	199 ^abC^	382 ^abB^	432 ^aA^	442 ^aA^	459 ^abA^				
	RY18	173 ^abC^	369 ^abB^	432 ^aA^	438 ^aA^	445 ^bA^				
	RY27	177 ^abC^	351 ^bB^	407 ^aA^	421 ^abA^	429 ^bcA^				
	RY36	163 ^bC^	297 ^cB^	367 ^bA^	380 ^bA^	403 ^cA^				
Peptide-N	RY0	177 ^aB^	237 ^aA^	245 ^A^	242 ^aA^	229 ^aA^	0.458	<0.001	<0.001	0.114
	RY9	165 ^abB^	232 ^abA^	240 ^A^	220 ^abA^	214 ^abA^				
	RY18	135 ^bcC^	226 ^abAB^	261 ^A^	227 ^abAB^	187 ^bB^				
	RY27	137 ^bcC^	222 ^abAB^	252 ^A^	218 ^abAB^	189 ^abBC^				
	RY36	120 ^cB^	179 ^bA^	219 ^A^	198 ^bA^	188 ^bA^				
FAA-N	RY0	27.0 ^cE^	134 ^aD^	166 ^aC^	194 ^aB^	225 ^aA^	0.723	<0.001	<0.001	<0.001
	RY9	29.6 ^bcE^	126 ^abD^	162 ^aC^	189 ^abB^	211 ^abA^				
	RY18	31.9 ^abD^	117 ^abC^	141 ^abC^	176 ^abB^	222 ^abA^				
	RY27	32.6 ^abD^	103 ^bcC^	126 ^bC^	168 ^bB^	199 ^bA^				
	RY36	34.9 ^aE^	91.9 ^cD^	118 ^bC^	143 ^cB^	172 ^cA^				
NH_3_-N	RY0	3.88 ^cD^	20.9 ^bC^	26.4 ^bB^	27.0 ^cB^	32.6 ^bA^	0.133	<0.001	<0.001	<0.001
	RY9	4.57 ^cC^	24.6 ^abB^	28.4 ^abB^	33.1 ^bA^	33.7 ^bA^				
	RY18	6.17 ^bD^	26.3 ^aC^	30.4 ^abB^	34.4 ^abA^	35.8 ^bA^				
	RY27	7.03 ^abC^	25.9 ^aB^	29.6 ^abB^	36.2 ^abA^	40.5 ^aA^				
	RY36	7.77 ^aE^	26.5 ^aD^	30.9 ^aC^	39.0 ^aB^	43.3 ^aA^				

Values in the same column (a–e) or in the same row (A–E) with different superscripts are significantly different (*p* < 0.05). NPN, non-protein nitrogen. Peptide-N, peptide nitrogen. FAA-N, free amino acid nitrogen. NH_3_-N, ammonia nitrogen. TN, total nitrogen. SEM, standard error of means. N = 3. T, effect of treatment. S, effect of storage days. T × S, effect of treatment and storage days interactions. RY0, 0% RY + 36% alfalfa. RY9, 9% RY + 27% alfalfa. RY18, 18% RY + 18% alfalfa. RY27, 27% RY + 9% alfalfa. RY36, 36% RY + 0% alfalfa.

**Table 7 animals-11-01273-t007:** Effect of paper mulberry replacement levels and storage days on the protein fraction of TMR silages (g/kg TN).

Items	Treatment	Storage Days					SEM	*p*-Value		
		0	7	14	28	56		T	S	T × S
Fraction A	RY0	208 ^aD^	392 ^aC^	438 ^aB^	463 ^aAB^	487 ^aA^	1.147	<0.001	<0.001	0.694
	RY9	199 ^abC^	382 ^abB^	431 ^aA^	442 ^aA^	459 ^abA^				
	RY18	173 ^abC^	369 ^abB^	432 ^aA^	438 ^aA^	445 ^bA^				
	RY27	177 ^abC^	351 ^bB^	407 ^aA^	421 ^abA^	429 ^bcA^				
	RY36	163 ^bC^	297 ^cB^	367 ^bA^	380 ^bA^	403 ^cA^				
True Protein	RY0	792 ^bA^	608 ^cB^	562 ^bC^	550 ^bCD^	526 ^cD^	1.147	<0.001	<0.001	0.694
	RY9	801 ^abA^	618 ^bcB^	569 ^bC^	558 ^bC^	542 ^bcC^				
	RY18	827 ^abA^	631 ^bcB^	578 ^bC^	562 ^abC^	555 ^bcC^				
	RY27	823 ^abA^	649 ^bB^	593 ^abC^	579 ^abC^	571 ^abC^				
	RY36	837 ^aA^	693 ^aB^	623 ^aC^	610 ^aC^	597 ^aC^				
Fraction B1	RY0	274 ^aA^	191 ^B^	194 ^B^	203 ^B^	199 ^B^	0.385	0.035	0.017	0.691
	RY9	245 ^abA^	194 ^AB^	189 ^AB^	175 ^B^	188 ^AB^				
	RY18	243 ^abA^	182 ^AB^	156 ^B^	177 ^B^	180 ^AB^				
	RY27	218 ^abA^	176 ^AB^	151 ^B^	180 ^AB^	165 ^B^				
	RY36	213 ^bA^	182 ^AB^	170 ^B^	170 ^B^	173 ^AB^				
Fraction B2	RY0	370 ^dA^	256 ^cB^	197 ^cC^	179 ^bCD^	150 ^cD^	1.061	<0.001	<0.001	0.840
	RY9	404 ^cdA^	264 ^cB^	235 ^bcBC^	201 ^bCD^	180 ^bcD^				
	RY18	433 ^bcA^	304 ^bB^	258 ^bBC^	221 ^bC^	204 ^abcC^				
	RY27	465 ^abA^	325 ^bB^	280 ^abBC^	230 ^abC^	242 ^abC^				
	RY36	482 ^aA^	360 ^aB^	330 ^aBC^	282 ^aCD^	270 ^aD^				
Fraction B3	RY0	103	111	119	120	125	1.109	0.200	0.167	0.944
	RY9	108	114	108	120	121				
	RY18	110	102	116	114	119				
	RY27	98.5	104	116	111	115				
	RY36	99.8	108	106	113	115				
Fraction C	RY0	45.4 ^B^	49.0 ^AB^	52.5 ^A^	49.0 ^AB^	52.0 ^A^	0.348	0.083	0.031	0.970
	RY9	43.5 ^B^	45.9 ^AB^	50.1 ^A^	46.2 ^AB^	52.3 ^A^				
	RY18	43.3 ^B^	45.5 ^AB^	49.3 ^AB^	47.8 ^AB^	51.1 ^A^				
	RY27	42.7 ^B^	44.8 ^AB^	47.4 ^AB^	45.5 ^AB^	49.0 ^A^				
	RY36	42.3 ^B^	43.1 ^AB^	47.5 ^A^	46.3 ^AB^	47.5 ^A^				

Values in the same column (a–e) or in the same row (A–E) with different superscripts are significantly different (*p* < 0.05). Fraction A, non-protein nitrogen. Fraction B1, buffer soluble protein. Fraction B2, neutral detergent soluble protein. Fraction B3, acid detergent soluble protein. Fraction C, acid detergent insoluble protein. TN, total nitrogen. SEM, standard error of means. N = 3. T, effect of treatment. S, effect of storage days. T×S, effect of treatment and storage days/interactions. RY0, 0% RY + 36% alfalfa. RY9, 9% RY + 27% alfalfa. RY18, 18% RY + 18% alfalfa. RY27, 27% RY + 9% alfalfa. RY36, 36% RY + 0% alfalfa.

**Table 8 animals-11-01273-t008:** Effect of paper mulberry replacement levels on in vitro digestibility of 56-day TMR silages.

Items	Treatment					SEM	*p*-Value		
	RY0	RY9	RY18	RY27	RY36		T	L	Q
In vitro dry matter digestibility (%)	73.6 ^a^	73.2 ^ab^	72.5 ^ab^	71.6 ^bc^	70.5 ^c^	0.219	0.024	0.003	0.723
In vitro crude protein digestibility (%)	73.0 ^a^	72.2 ^ab^	71.5 ^ab^	70.6 ^bc^	69.5 ^c^	0.675	0.010	0.004	0.281
In vitro neutral detergent fiber digestibility (%)	56.7	56.3	55.8	55.3	54.9	0.687	0.074	0.118	0.612

Values in the same row (a–c) with different superscripts are significantly different (*p* < 0.05). T, treatment effects. L, linear effects. Q, quadratic effects. SEM, standard error of means. RY0, 0% RY + 36% alfalfa. RY9, 9% RY + 27% alfalfa. RY18, 18% RY + 18% alfalfa. RY27, 27% RY + 9% alfalfa. RY36, 36% RY + 0% alfalfa.

## Data Availability

The data presented in this study are available on request from the corresponding author.

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
