# Peer review of "Replacing Alfalfa with Paper Mulberry in Total Mixed Ration Silages: Effects on Ensiling Characteristics, Protein Degradation, and In Vitro Digestibility"

_animals, 2021, doi:10.3390/ani11051273_

Round 1

Reviewer 1 Report

The manuscript was largely improved in the language and content exposition. Statistics is now well presented and very clear. Also, the tables were corrected and simplified.

I really appreciate the author's revisions and corrections.

I only suggest revising the form of the manuscript, if it is alike the journal given form, especially for tables, text size, and spatial distribution.

Author Response

Point 1: The manuscript was largely improved in the language and content exposition. Statistics is now well presented and very clear. Also, the tables were corrected and simplified. I really appreciate the author's revisions and corrections.

Response 1: Thank you very much for the comments.

Point 2: I only suggest revising the form of the manuscript, if it is alike the journal given form, especially for tables, text size, and spatial distribution.

Response 2: We appreciate the valuable suggestions. We checked form of manuscript according to the journal given form. The line spacing of manuscript title was corrected to 12 pounds. Please see [Page#1; Lines 2 to 4]. “37°C” was corrected to “37 °C” and “30°C” was corrected to “30 °C”. Please see [Page#3; Lines 121 and Line 124]. The title of table 4 was moved to next page. Please see [Page#7; Line 235]. The title of table 7 was moved to next page. Please see [Page#9; Line 286]. In results section, we added the point to 3.7 (in vitro digestibility of 56-day TMR silages) section. Please see [Page#9; Line 293]. In author contributions and reference section, short front spacing was corrected to 12 pounds. Please see [Page#12; Line 450] and [Page#13; Line 472].  

Reviewer 2 Report

Improved work compared to the initial version, but still with some suggested changes that have not been modified.

Global comments:

  • Please it would be interesting to make an index of abbreviations, it would serve to better understand the text.
  • What about amino acids? It would be interesting to add this information both in the introduction and above all in the discussion, since the CP is not important but the AA. If the analysis has not been carried out in the laboratory, the ideal would be even to do it with theoretical data.

Specific comments:

The data was not checked for normality? kurtosis? Can anova be applied? If it has been done it must be specified, if it has not been done it should be done.

(Table 1) No standard error? This should wear.

¿Do you think that the number of samples is representative?

Author Response

Comments and Suggestions for Authors (Global comments)

Point 1: Improved work compared to the initial version, but still with some suggested changes that have not been modified.

Response 1: Thanks for the comments.

Point 2: Please it would be interesting to make an index of abbreviations, it would serve to better understand the text.

Response 2: We appreciate the valuable suggestion. Acronym index table was added to better understand the manuscript. Please see [Page#13; Lines 471].

Point 3: What about amino acids? It would be interesting to add this information both in the introduction and above all in the discussion, since the CP is not important but the AA. If the analysis has not been carried out in the laboratory, the ideal would be even to do it with theoretical data.

Response 3: We appreciate the valuable suggestion. Some information was added to the manuscript. Please see [Page#11; Lines 384 to 389].

Amino acid nutrition of the diet can influence animal performance, and balanced amino acid nutrition of the diet is essential for maximizing nutrient utilization and productivity of dairy and beef cattle. Quantitative evaluation of amino acid composition of the diet offers important reference to meeting the nutritional needs of dairy cows.

The aim of this study was preliminary evaluation on the dynamics of fermentation characteristics, chemical composition, protein degradation and in vitro digestibility of total mixed ration (TMR) silages when paper mulberry (Broussonetia papyrifera L., RY) was substituted for alfalfa at different ratios. Our results showed that RY substitution had no adverse effect on fermentation quality and nutritional composition of TMR silages. After ensiling, the contents of non-protein nitrogen, peptide nitrogen and free amino acid nitrogen for RY18 silages was lower than that of the control, but the buffer soluble protein and acid detergent soluble protein fractions was not different among the all silages. Digestibility of dry matter and crude protein for RY27 and RY36 silages was lower than the control, but there was no difference between control and RY18 silages. Therefore, it appears that TMR silages produced with RY to alfalfa ratio of 18:18 was suitable as a result of balance the silage quality, protein degradation and in vitro digestibility.

Amino acid levels (including essential amino acid and non-essential amino acid) are not determined but the total free amino acid nitrogen in this experiment. Further quantitative evaluation of amino acid degradation in TMR silages is essential to add the more information on accurate animal feeding and production when RY is introduced. Thanks for pointing out the direction of our future research. Buddle of thanks for the valuable suggestion.

Specific comments:

Point 4: The data was not checked for normality? kurtosis? Can ANOVA be applied? If it has been done it must be specified, if it has not been done it should be done.

Response 4: Thanks for the comments. The sentence “Prior to statistical analysis, all data is subjected to Homogeneity test of variance. The normality was tested with Shapiro–Wilk test. All data can be analyzed by ANOVA” was added to statistical analysis section. Please see [Page#4; Lines 168 to 170]. The analytical results of were as follows:

Table 1. Homogeneity test of variance of the chemical composition and in vitro digestibility.

Items

levene statistics

df1

df2

sig.

Fresh materials

Dry matter

0.902

4

10

0.499

Crude protein

1.378

4

10

0.309

Neutral detergent fiber

0.312

4

10

0.864

Acid detergent fiber

3.074

4

10

0.074

Acid detergent lignin

2.132

4

10

0.151

Water soluble carbohydrates

0.413

4

10

0.796

Buffer capacity

3.245

4

10

0.066

Condensed tannin

0.616

4

10

0.661

Hydrolysable tannin

0.604

4

10

0.668

After ensiling

Dry matter

2.302

4

10

0.130

Crude protein

1.801

4

10

0.205

Neutral detergent fiber

0.148

4

10

0.960

Acid detergent fiber

0.919

4

10

0.490

Acid detergent lignin

2.322

4

10

0.156

Water soluble carbohydrates

2.328

4

10

0.127

Buffer capacity

0.965

4

10

0.468

Condensed tannin

1.714

4

10

0.223

Hydrolysable tannin

0.749

4

10

0.580

In vitro dry matter digestibility

1.808

4

10

0.204

In vitro crude protein digestibility

0.984

4

10

0.459

In vitro neural detergent fiber digestibility

1.526

4

10

0.267

 The significance value of chemical composition (including fresh materials and ensiling after 56 days) and in vitro digestibility (ensiling after 56 days) parameters was more than 0.05 (Table 1). Therefore, this result is consistent with the homogeneity test of variance.

Table 2. Normality test of the chemical composition and in vitro digestibility.

Items

Kolmogorov-Smirnov

Shapiro-Wilk

statistics

df

sig.

statistics

df

sig.

Fresh materials

Dry matter

0.144

15

0.200

0.937

15

0.345

Crude protein

0.129

15

0.200

0.955

15

0.611

Neutral detergent fiber

0.088

15

0.200

0.973

15

0.896

Acid detergent fiber

0.142

15

0.200

0.947

15

0.486

Acid detergent lignin

0.107

15

0.200

0.975

15

0.925

Water soluble carbohydrates

0.140

15

0.200

0.969

15

0.836

Buffer capacity

0.174

15

0.200

0.917

15

0.172

Condensed tannin

0.155

15

0.200

0.915

15

0.161

Hydrolysable tannin

0.171

15

0.200

0.912

15

0.146

After ensiling

Dry matter

0.112

15

0.200

0.959

15

0.670

Crude protein

0.152

15

0.200

0.938

15

0.359

Neutral detergent fiber

0.142

15

0.200

0.971

15

0.878

Acid detergent fiber

0.167

15

0.200

0.933

15

0.303

Acid detergent lignin

0.176

15

0.200

0.962

15

0.719

Water soluble carbohydrates

0.180

15

0.200

0.949

15

0.507

Buffer capacity

0.114

15

0.200

0.977

15

0.949

Condensed tannin

0.169

15

0.200

0.911

15

0.141

Hydrolysable tannin

0.141

15

0.200

0.912

15

0.145

In vitro dry matter digestibility

0.131

15

0.200

0.974

15

0.908

In vitro crude protein digestibility

0.144

15

0.200

0.946

15

0.468

In vitro neural detergent fiber digestibility

0.177

15

0.200

0.927

15

0.245

The significance value of chemical composition (including fresh materials and ensiling after 56 days) and in vitro digestibility (ensiling after 56 days) parameters was more than 0.05 according to Shapiro-Wilk test (Table 2). Therefore, this result is consistent with normality test.

Table 3. Homogeneity test of variance of the fermentation quality, protein and microbial composition.

Items

levene statistics

df1

df2

sig.

0 day

Non-protein nitrogen

1.213

4

10

0.365

Peptide nitrogen

1.314

4

10

0.329

Free amino acid nitrogen

0.129

4

10

0.968

Ammonia nitrogen

2.049

4

10

0.163

Neutral detergent soluble protein

1.009

4

10

0.448

Acid detergent soluble protein

0.811

4

10

0.546

Acid detergent insoluble protein

3.052

4

10

0.069

7 day

pH

1.467

4

10

0.283

Latic acid

1.027

4

10

0.171

Acetic acid

2.203

4

10

0.142

Propionic acid

1.081

4

10

0.516

Non-protein nitrogen

0.650

4

10

0.640

Peptide nitrogen

2.473

4

10

0.112

Free amino acid nitrogen

1.012

4

10

0.446

Ammonia nitrogen

1.121

4

10

0.400

Neutral detergent soluble protein

2.218

4

10

0.140

Acid detergent soluble protein

0.499

4

10

0.737

Acid detergent insoluble protein

1.888

4

10

0.189

Lactic acid bacteria

1.466

4

10

0.168

Aerobic bacteria

1.059

4

10

0.144

14 day

pH

1.412

4

10

0.299

Latic acid

1.725

4

10

0.221

Acetic acid

1.204

4

10

0.368

Propionic acid

0.355

4

10

0.835

Non-protein nitrogen

0.981

4

10

0.460

Peptide nitrogen

1.577

4

10

0.254

Free amino acid nitrogen

1.942

4

10

0.136

Ammonia nitrogen

3.076

4

10

0.055

Neutral detergent soluble protein

0.469

4

10

0.873

Acid detergent soluble protein

0.423

4

10

0.789

Acid detergent insoluble protein

1.150

4

10

0.388

Lactic acid bacteria

1.759

4

10

0.211

Aerobic bacteria

1.013

4

10

0.279

28 day

pH

2.794

4

10

0.085

Latic acid

1.129

4

10

0.397

Acetic acid

2.013

4

10

0.169

Propionic acid

0.225

4

10

0.918

Non-protein nitrogen

0.761

4

10

0.574

Peptide nitrogen

2.088

4

10

0.157

Free amino acid nitrogen

1.116

4

10

0.172

Ammonia nitrogen

2.000

4

10

0.171

Neutral detergent soluble protein

2.141

4

10

0.116

Acid detergent soluble protein

1.397

4

10

0.303

Acid detergent insoluble protein

1.083

4

10

0.415

Lactic acid bacteria

0.079

4

10

0.277

Aerobic bacteria

1.012

4

10

0.204

56 day

pH

2.933

4

10

0.076

Latic acid

0.056

4

10

0.993

Acetic acid

3.291

4

10

0.058

Propionic acid

0.974

4

10

0.463

Non-protein nitrogen

0.673

4

10

0.626

Peptide nitrogen

1.651

4

10

0.237

Free amino acid nitrogen

1.467

4

10

0.283

Ammonia nitrogen

1.910

4

10

0.185

Neutral detergent soluble protein

1.185

4

10

0.375

Acid detergent soluble protein

1.094

4

10

0.411

Acid detergent insoluble protein

1.494

4

10

0.276

Lactic acid bacteria

1.741

4

10

0.169

Aerobic bacteria

1.218

4

10

0.174

The significance value of fermentation quality, protein and microbial composition during ensiling was more than 0.05 (Table 3). Therefore, this result is consistent with the homogeneity test of variance.

Table 4. Normality test of the fermentation quality, protein and microbial composition.

Items

Kolmogorov-Smirnov

Shapiro-Wilk

statistics

df

sig.

statistics

df

sig.

0 day

Non-protein nitrogen

0.165

15

0.200

0.924

15

0.225

Peptide nitrogen

0.181

15

0.197

0.910

15

0.136

Free amino acid nitrogen

0.104

15

0.200

0.979

15

0.963

Ammonia nitrogen

0.175

15

0.200

0.925

15

0.233

Neutral detergent soluble protein

0.177

15

0.200

0.943

15

0.420

Acid detergent soluble protein

0.107

15

0.200

0.989

15

0.999

Acid detergent insoluble protein

0.174

15

0.200

0.930

15

0.274

7 day

pH

0.179

15

0.200

0.918

15

0.181

Latic acid

0.178

15

0.200

0.949

15

0.507

Acetic acid

0.169

15

0.200

0.904

15

0.110

Propionic acid

0.164

15

0.200

0.936

15

0.337

Non-protein nitrogen

0.205

15

0.089

0.884

15

0.054

Peptide nitrogen

0.169

15

0.200

0.876

15

0.041

Free amino acid nitrogen

0.104

15

0.200

0.938

15

0.356

Ammonia nitrogen

0.231

15

0.030

0.890

15

0.068

Neutral detergent soluble protein

0.167

15

0.200

0.942

15

0.413

Acid detergent soluble protein

0.134

15

0.200

0.936

15

0.337

Acid detergent insoluble protein

0.203

15

0.096

0.902

15

0.101

Lactic acid bacteria

0.153

15

0.200

0.907

15

0.216

Aerobic bacteria

0.107

15

0.200

0.911

15

0.458

14 day

pH

0.207

15

0.083

0.930

15

0.272

Latic acid

0.096

15

0.200

0.965

15

0.778

Acetic acid

0.149

15

0.200

0.904

15

0.111

Propionic acid

0.138

15

0.200

0.935

15

0.319

Non-protein nitrogen

0.217

15

0.056

0.898

15

0.062

Peptide nitrogen

0.162

15

0.200

0.936

15

0.340

Free amino acid nitrogen

0.189

15

0.200

0.969

15

0.373

Ammonia nitrogen

0.153

15

0.200

0.939

15

0.371

Neutral detergent soluble protein

0.168

15

0.200

0.930

15

0.275

Acid detergent soluble protein

0.176

15

0.200

0.931

15

0.280

Acid detergent insoluble protein

0.138

15

0.200

0.939

15

0.373

Lactic acid bacteria

0.203

15

0.200

0.954

15

0.501

Aerobic bacteria

0.141

15

0.200

0.918

15

0.419

28 day

pH

0.165

15

0.200

0.970

15

0.865

Latic acid

0.142

15

0.200

0.959

15

0.679

Acetic acid

0.142

15

0.200

0.925

15

0.228

Propionic acid

0.152

15

0.200

0.963

15

0.741

Non-protein nitrogen

0.123

15

0.200

0.949

15

0.511

Peptide nitrogen

0.144

15

0.200

0.957

15

0.640

Free amino acid nitrogen

0.250

15

0.200

0.947

15

0.616

Ammonia nitrogen

0.167

15

0.200

0.949

15

0.507

Neutral detergent soluble protein

0.162

15

0.200

0.926

15

0.235

Acid detergent soluble protein

0.136

15

0.200

0.952

15

0.551

Acid detergent insoluble protein

0.105

15

0.200

0.942

15

0.413

Lactic acid bacteria

0.146

15

0.200

0.962

15

0.519

Aerobic bacteria

0.139

15

0.200

0.927

15

0.647

56 day

pH

0.207

15

0.082

0.908

15

0.125

Latic acid

0.148

15

0.200

0.944

15

0.442

Acetic acid

0.132

15

0.200

0.910

15

0.138

Propionic acid

0.174

15

0.200

0.925

15

0.232

Non-protein nitrogen

0.103

15

0.200

0.976

15

0.938

Peptide nitrogen

0.159

15

0.200

0.965

15

0.779

Free amino acid nitrogen

0.160

15

0.200

0.944

15

0.432

Ammonia nitrogen

0.211

15

0.071

0.889

15

0.057

Neutral detergent soluble protein

0.132

15

0.200

0.951

15

0.542

Acid detergent soluble protein

0.124

15

0.200

0.973

15

0.905

Acid detergent insoluble protein

0.170

15

0.200

0.945

15

0.445

Lactic acid bacteria

0.181

15

0.200

0.933

15

0.471

Aerobic bacteria

0.142

15

0.200

0.915

15

0.508

The significance value of the fermentation quality, protein and microbial composition during ensiling was more than 0.05 (Table 4). Therefore, this result is consistent with normality test.

Point 5: (Table 1) No standard error? This should wear.

Response 5: Thanks for point out. Standard error of means was added to the table 1. Please see [Page#5; Lines 194 and 197].

Point 6: Do you think that the number of samples is representative?

Response 6: Thanks for the comments. In this study, TMR were made with alfalfa and RY mixtures (36.0%), maize meal (35.0%), oat grass (10.0%), soybean meal (7.5%), brewers’ grain (5.0%), wheat bran (5.0%), pre-mix (1.0%), and salt (0.5%) on dry matter basis, respectively. The alfalfa and RY mixtures were made in the following ratios of dry matter: 36:0 (RY0), 27:9 (RY9), 18:18 (RY18), 9:27 (RY27) and 0:36 (RY36). The experimental diets were formulated according to Chinese Feeding Standard for Dairy Cattle [1]. Five storage periods (0 d, 7 d, 14 d, 28 d and 56 d) were choose basing on the critical period for silage. For each treatment, three replicates of silage samples were prepared and a total of 75 mini silos were obtained for this experiment. After 0, 7, 14, 28 and 56 days of ensiling respectively, triplicate samples of silage from each treatment were opened. Numerous studies [2-5] also set three triplicates of each treatment to evaluate the feed value for silage.

 According to previous studies [6-7], a total of one hundred and thirty-five filter bags (five treatments × three individual samples × three filter bags per sample × three runs) were prepared for in vitro digestibility analysis in this study, thus each treatment contained nine replicates.

Reference

[1] Association JGFFS. Guide book for quality evaluation of forage. Tokyo, Japan. 1994, 82-87.

[2] Gao R, Luo Y, Xu S, Wang M, Sun Z, Wang L, Yu, Z. Effects of replacing ensiled-alfalfa with fresh-alfalfa on dynamic fermentation characteristics, chemical compositions, and protein fractions in fermented total mixed ration with different additives. Animals, 2021, 11, 572.

[3] Xie Y, Xu S, Li W, Bao J, Jia T, Yu Z. Effects of the application of Lactobacillus plantarum Inoculant and potassium sorbate on the fermentation quality, in vitro digestibility and aerobic stability of total mixed ration silage based on alfalfa silage. Animals, 2020, 12, 2229.

[4] Du Z, Sun L, Chen C, Lin J, Yang F, Cai Y. Exploring microbial community structure and metabolic gene clusters during silage fermentation of paper mulberry, a high-protein woody plant. Animal Feed Science and Technology, 2021, 275, 114766.

[5] Tian P, Niu D, Zuo S, Jiang D, Li R, Xu C. Vitamin A and E in the total mixed ration as influenced by ensiling and the type of herbage. Science of The Total Environment, 2020, 746, 141239.

[6] Guo L, Yao D, Li D, Lin Y, Bureenok S, Ni K, Yang F. Effects of lactic acid bacteria isolated from rumen fluid and feces of dairy cows on fermentation quality, microbial community, and in vitro digestibility of alfalfa silage. Frontiers in microbiology, 2020, 10, 2998. 

[7] Lei C, Dong Z, Li J, Shao T. Ensiling characteristics, in vitro rumen fermentation, microbial communities and aerobic stability of low‐dry matter silages produced with sweet sorghum and alfalfa mixtures. Journal of the Science of Food and Agriculture, 2019, 99, 2140-2151.

[5] Tian P, Niu D, Zuo S, Jiang D, Li R, Xu C. Vitamin A and E in the total mixed ration as influenced by ensiling and the type of herbage. Science of The Total Environment, 2020, 746, 141239.

[6] Guo L, Yao D, Li D, Lin Y, Bureenok S, Ni K, Yang F. Effects of lactic acid bacteria isolated from rumen fluid and feces of dairy cows on fermentation quality, microbial community, and in vitro digestibility of alfalfa silage. Frontiers in microbiology, 2020, 10, 2998. 

[7] Lei C, Dong Z, Li J, Shao T. Ensiling characteristics, in vitro rumen fermentation, microbial communities and aerobic stability of low‐dry matter silages produced with sweet sorghum and alfalfa mixtures. Journal of the Science of Food and Agriculture, 2019, 9

This manuscript is a resubmission of an earlier submission. The following is a list of the peer review reports and author responses from that submission.

Round 1

Reviewer 1 Report

The research presented cover a useful topic

It provide the ground information to continue in vivo evaluation of novel feeds

The research seems to be adequately performed however there numerous details missing in the M&M section which need to be clarify to confirm that the experimental design and statistical analysis are valid.

Methodological procedure details are missing for some variables

Otherwise the discussion seems appropriated and only need to moderate the conclusion.

(see attached file)

However, given the missing methodological details my recommendation is conservative until further details are provided by authors in a revised version.

Author Response

Dear reviewer:

We corrected the manuscript carefully according to the comments. Please see the attachment.

Reviewer 2 Report

Dear Authors,

this manuscript is aimed at an interesting purpose, with valuable results. However, it needs to be written again. The references, especially in the introduction, are not responding to the sentences. Acronyms are chaotic and they do not be the same as the tables. Statistic analysis needs references and model formulas with variable factors as reported in the tables. The chemical analysis of the forage pre-fermentation is reported without statistics, nevertheless, is commented as different. All the experimental design of the in vitro trial is missing. No mention of numbers of reactors per treatment and about fermentation conditions.

Author Response

We revised the manuscript carefully basing on the comments, please see the attachment. We hope that revised manuscript will be good quality than before.

Kind regards, 

Rongrong Li

Reviewer 3 Report

Global comments:

  • Please it would be interesting to make an index of abbreviations, it would serve to better understand the text.
  • General comment about tables: The tables are in different pages, please wathc the format. For example punctuation.
  • What about amino acids? It would be interesting to add this information both in the introduction and above all in the discussion, since the CP is not important but the AA. If the analysis has not been carried out in the laboratory, the ideal would be even to do it with theoretical data.
  • I think a more in-depth study should be done on the nutrients that cause these changes between raw materials to enrich the discussion.

Specific comments:

L23. all Latin names must be in italics

L26. The ratios is not well explicated. Is alfalfa:mulberry? Please explain it better.

L28. It has a relationship between in vitro and in vivo?

L26. In vitro in italics. Please see all names.

L28. Please see comments on L 23.             

L70. Please add a space.

L137. The data was not checked for normality? kurtosis? Can anova be applied? If it has been done it must be specified, if it has not been done it should be done.

L155. Please explain DM and FM.

L155. TN is not refered in the Table 1.

L168. What is TMR? You should explain it.

L201. Table is in two pages.

L215. Point after tittle  (among many).

L310. Is the only moment when you talk about amino acid. You should expand the information on this side a lot, since even the title itself talks about protein.

L349. Please, explain in more depth the in vivo correlation in vitro, as it is what really matters in production.

Author Response

We have answered the questions and corrected the mistakes in manuscripts who reviewer raised. Please download the attachment.

Kind regards,

Rongrong Li